# Assessment approaches and methods in physiotherapy education: A scoping review protocol

Katie L. Kowalski[1]*, Andrews Tawiah[1], Erin Miller[1], Maren Goodman[2], Greg Alcock[1], Heather Gillis[1]

1 School of Physical Therapy, Western University, London, Ontario, Canada, 2 Western Libraries, Western University, London, Ontario, Canada

* kkowals7@uwo.ca

## Abstract

### Introduction

Assessment in physiotherapy education is essential to ensuring graduates meet professional standards, demonstrate competence and are prepared to provide high-quality, patient-centered care. Assessment approaches (principles guiding assessment design, sequence and implementation) and methods (techniques used to evaluate performance) should be constructively aligned with curriculum content and learning outcomes, and assess the breadth of competencies required for practice. While individual studies describe assessment approaches and methods in physiotherapy education, there is no comprehensive synthesis or strategic analysis of their strengths, weaknesses, opportunities and threats.

### Objectives

To 1) identify and characterize assessment approaches and methods used in physiotherapy education; 2) analyze the strengths, weaknesses, opportunities and threats of assessment approaches and methods used in physiotherapy education.

### Methods

This scoping review will follow Joanna Briggs Institute guidance and report according to Preferred Reporting Items for Systematic review and Meta-analysis extensions for Protocols and Scoping Reviews. The protocol is registered with Open Science Framework. Searches will be performed in Medline, CINAHL, EMBASE, ProQuest Education Research Index, Scopus, PsycINFO, and relevant grey literature from inception until July 23, 2025. Eligible studies will report primary research on assessment approaches or methods in the academic curriculum of university-based clinical physiotherapy education programs. Studies focusing only on clinical education,

**Data availability statement:** No datasets were generated or analysed during the current study. All relevant data from this study will be made available upon study completion.

**Funding:** The author(s) received no specific funding for this work.

**Competing interests:** The authors have declared that no competing interests exist.

reviews, opinion pieces will be excluded. Two reviewers will independently screen studies and extract data using a standardized tool. A narrative synthesis will characterize assessment approaches and methods, categorizing approaches by guiding principles and methods by type. Methods will be mapped to Miller's pyramid of clinical competence. A directed content analysis will identify the strengths, weaknesses, opportunities and threats influencing assessments in physiotherapy education.

## Conclusion

This review will provide a comprehensive synthesis and strategic analysis of assessments in physiotherapy education. Findings will inform evidence-based assessment practices that support optimal student learning, experiences and readiness for contemporary practice.

## Introduction

Physiotherapy is an internationally recognized regulated health profession that plays a vital role within health systems to optimize movement and functional abilities [1,2]. Physiotherapy educational programs at all levels (e.g., entry-level, postgraduate level) prepare students for the responsibilities of independent and autonomous practice through rigorous academic curricula, ensuring graduates meet professional and societal expectations for effective, safe and ethical patient-centered care [3,4]. Within educational programs, assessment plays a critical role in supporting student learning, guiding academic progression and contributing to the development of competent physiotherapy practitioners. Through constructive alignment, assessment links curriculum content with the knowledge, skills and attributes learners are expected to demonstrate [5]. Assessment is therefore essential in ensuring physiotherapy graduates meet professional standards, demonstrate competence and are prepared to provide high quality, patient centered care.

The World Physiotherapy Education Framework advocates for an assessment approach that is comprehensive and fair, using a range of assessment methods aligned with intended learning outcomes [4]. Assessment approach refers to the broad set of principles that guide how assessment is designed, sequenced and implemented within an educational program [6,7]. These principles are shaped by the program's underpinning pedagogy and inform assessment purpose, frequency and interpretation of outcomes. For example, a programmatic assessment approach is built on a range of principles, including the use of substantial formative feedback, multiple assessments to inform high-stakes pass/fail decisions, and the progressive development of learner agency and accountability for learning [8]. In contrast, assessment methods are the specific techniques used to collect information about a student's performance at a specific point in time [6]. In health professions education, four major assessment methods are commonly described: written tests, oral examinations, performance tests (e.g., objective structured clinical examination) and workplace-based assessment (e.g., clinical observation) [9]. Miller's pyramid of

clinical competence provides a structure for organizing assessment methods by the level of learning they target, ranging from assessing knowledge ("knows") to action in a clinical environment ("does") [10]. Owing to the breadth and complexity of competencies that students must develop in physiotherapy education, a range of assessment methods is necessary to evaluate multiple domains of practice and demonstrate achievement of professional standards [4].

Existing literature on assessment in health professions education highlights an evolution towards student-centered approaches that prepare students to practice in complex and dynamic clinical environments [11]. Evidence syntheses have explored approaches such as programmatic assessment and authentic assessment, emphasizing both their potential to enhance learning and the challenges associated with implementation. For example, a review comprising primarily of medical education found programmatic assessment can lead to meaningful, student-driven learning, but challenges exist, such as a lack of shared understanding between educators and students about assessment purpose and use [12]. Scoping review evidence also identified programmatic assessment as valuable to support student learning, but resource-intensive and therefore potentially challenging to implement in low-resource educational contexts [6]. In nursing education, authentic assessment can lead to deep learning by engaging students in real-world activities, but different assessment methods can affect student engagement and assessment outcomes [13].

In physiotherapy education, existing literature consists of individual studies exploring assessment approaches and methods. For example, authentic assessment has been used to support learning in exercise prescription by encouraging application of knowledge in realistic context, with clear expectations being critical to optimize learning outcomes [14]. Programmatic assessment in physiotherapy shows promise in supporting flexible, self-directed and collaborative learning, with challenges such as how to best guide students through individualized learning pathways [15]. Other studies have focused on assessment tool development and validation for clinical procedural skills. For example, establishing the structural validity of the Assessment of Procedural Skills in Physiotherapy Education tool, providing evidence for its use in evaluating physiotherapy procedural skills within the academic curriculum [16]. While individual studies offer valuable insight, there is no comprehensive synthesis of assessment approaches and methods in physiotherapy education, or analysis of their associated strengths, weaknesses, opportunities, and threats (SWOT). A SWOT analysis enables a strategic evaluation of internal features (e.g., feasibility) and external influences (e.g., advances in technology) that shape assessments in physiotherapy education, offering practical insights into how assessments can be refined to optimize educational quality.

### Objectives

1. To identify and characterize the assessment approaches and methods used in physiotherapy education.

2. To analyze the strengths, weaknesses, opportunities and threats of assessment approaches and methods used in physiotherapy education.

## Methods

### Design

A scoping review was selected as the study design owing to the broad, descriptive, and exploratory nature of the research objectives [17]. This scoping review protocol is designed according to JBI guidance, and reporting aligns with Preferred Reporting Items for Systematic review and Meta-analysis – Protocols [18–20]. Reporting of the definitive scoping review will align with the PRISMA Scoping Review extension [21]. The protocol is registered with Open Science Framework (https://osf.io/g7tx3), which will be updated with any protocol amendments.

### Scoping review team

In accordance with best practice recommendations for conducting scoping reviews [18], the research team includes expertise in physiotherapy education, scoping review methodology and information science. The team includes several

faculty members who are actively involved in teaching and assessment within entry-to-practice and postgraduate physiotherapy education programs, bringing both content expertise and lived experience as users of evidence-informed assessment strategies. A Teaching and Learning Librarian with expertise in information retrieval and search strategy development is included as part of the team. All research team members have prior experience conducting scoping reviews and are familiar with JBI methodology.

## Eligibility criteria

**Population.** Studies will be included if they involve students enrolled in a physiotherapy education program at any level (e.g., undergraduate, graduate, postgraduate). Studies that focus on interprofessional education will be included if data specific to physiotherapy can be extracted.

**Concept.** Studies must describe assessment approaches or methods in physiotherapy academic curriculum. Assessment approach is defined as a set of principles that guide assessment design, sequence and implementation in an educational program (e.g., programmatic assessment), while assessment method refers to a specific technique used to collect information on the performance of a student (e.g., objective structured clinical examination) [6]. Studies will be excluded if assessment approaches or methods are examined only to evaluate delivery and outcomes of educational interventions. Studies focusing on assessment approaches or methods in clinical education will also be excluded.

**Context.** Assessments must occur within a university-based clinical physiotherapy education program. Studies conducted in any country will be eligible for inclusion.

**Additional criteria.** Any study design reporting primary research will be eligible. Opinion pieces (e.g., Editorials), commentaries, response to reviewers, and reviews will be excluded. If a relevant review is identified, we will search its included studies for potentially relevant primary research studies to include. Published abstracts and conference proceedings will be excluded; however, if any are identified that appear relevant, we will search for the full-text paper to include. Studies published in any language will be included. In alignment with best practice recommendations [22,23], if a study is written in a language other than English, Google Translate will be used to generate an initial translation (≥85% translation accuracy across a range of languages [24]), which will then be reviewed and edited as needed by someone proficient in the language the study is written to ensure accuracy of translation. Method of translation for each included study written in a language other than English will be reported for transparency.

## Information sources

Key electronic databases will be searched from inception until July 23, 2025, including Medline (Ovid), CINAHL (EBSCO), EMBASE (Ovid), ProQuest Education Research Index, Scopus, PsycINFO (Ovid). Grey literature will be searched using ProQuest Dissertations and Theses and an advanced Google search limited to the first 200 results, ensuring a comprehensive and pragmatic search. If relevant reviews are identified, the reference list will be screened to identify additional potentially eligible studies.

## Search strategy

The search strategy will be developed in collaboration with a Teaching and Learning Librarian. The search was developed in Medline (S1 Appendix) and then adapted for other databases. Search terms are structured around three constructs (physiotherapy, assessment, education) and informed by relevant systematic reviews in health professions education assessment [6,25]. The search strategy will be peer-reviewed by a Librarian independent of the research team using the Peer Review of Electronic Search Strategies checklist [26]. No revisions to the search strategy were suggested or required (S2 Appendix).

## Selection of sources of evidence

Citations retrieved by the search strategy will be imported and stored in Covidence, a web-based platform that enables a collaborative and streamlined approach to evidence synthesis research projects. Covidence will automatically identify and remove duplicate citations. Study eligibility screening, including title/abstract and full text screening, will also occur in Covidence.

Study eligibility screening will be performed independently by two reviewers. Title and abstracts will be screened against eligibility criteria using the following criteria: eligible, not eligible, may be eligible. Full texts will be retrieved for studies in which both reviewers agree the study is, or may be, eligible for inclusion. A study will be included when both reviewers independently determine the full text study meets eligibility criteria. To enhance consistency between reviewers, pilot screening will be conducted, including 50 citations during title and abstract screening and five texts during full-text screening. Pilot eligibility screening will be assessed based on an agreement of 80% or higher between reviewers at each stage. If agreement is less than 80%, eligibility criteria will be reviewed and clarified before proceeding with screening. Disagreements about eligibility at the title/abstract and full text screening stages will first be discussed between the two reviewers, and if consensus is not achieved, a third reviewer will mediate. At the time of protocol manuscript submission, title/abstract screening had commenced and less than 5% of citations had been screened.

## Data charting process

The research team will develop a standardized data extraction tool, in accordance with best practice recommendations for scoping reviews [27]. Two reviewers will independently extract data in Microsoft Excel, beginning with a pilot stage of five articles, ensuring the tool is fit for purpose and consistent data extraction. Discrepancies in data extracted will be resolved through discussion between reviewers and if consensus is not achieved, a third reviewer will mediate. If data are unclear or not presented in eligible studies, corresponding authors will be contacted with a request for additional information. As needed, one follow-up email will be sent two weeks later. It is anticipated data charting will be complete by end of January 2026.

## Data items

Data to be extracted from eligible studies is summarized in Table 1.

## Synthesis of results

Methodological features (e.g., study design, sample size) will be narratively described across included studies The synthesis of results will combine narrative synthesis and directed content analysis to address objectives 1 and 2 [27].

For objective 1, a narrative synthesis will be conducted to characterize the assessment approaches and methods used in physiotherapy education. The identified assessment approaches will be categorized according to their underlying guiding principles. The commonalities and differences between the various approaches will be examined. Assessment methods will be systematically grouped by type, creating a comprehensive taxonomy of evaluation techniques employed in physiotherapy education. The frequency and context of method usage will be described in detail, providing insights into which assessment methods are most commonly used across different educational settings and course levels. The identified assessment methods will be mapped onto Miller's pyramid of clinical competence to examine how different evaluation techniques address the progressive levels of learning and competency development [10]. This mapping will categorize methods according to whether they primarily assess foundational knowledge ("knows"), applied knowledge and problem-solving ("knows how"), demonstration of skills in controlled settings ("shows how"), or performance in real-world clinical practice ("does").

**Table 1. Data to be extracted from eligible studies.**

| Study characteristics | • Authors<br>• Title<br>• Year of publication<br>• Country of study | • Language of publication<br>• Study design and methodology<br>• Study objectives |
|---|---|---|
| Physiotherapy educational program | • Level of education<br>• Characteristics of educational program/ course | |
| Student characteristics | • Sample size<br>• Demographics (age, sex) | • Year/ Stage of program |
| Assessment | Characteristics of approach<br>• Name or type of approach<br>• Description of principles, goals or values that guide approach (e.g., formative focus)<br>• Purpose of approach (e.g., for learning, assess competence)<br>• Structure of approach (e.g., organization, frequency, timing)<br>• Strengths<br>• Limitations | Characteristics of methods<br>• Type of method used (e.g., Objective structured clinical examination)<br>• Description of method (e.g., setting, format)<br>• Competency being assessed<br>• Purpose of method (e.g., summative)<br>• Delivery mode (e.g., in-person, online)<br>• Assessment tool<br>• Strengths<br>• Limitations |

For objective 2, a directed content analysis using the SWOT (Strengths, Weaknesses, Opportunities, and Threats) framework will be employed to provide a strategic analysis of assessment approaches and methods in physiotherapy education [28–30]. The elements in the SWOT framework will be used as the initial coding categories. SWOT will be conceptualized as follows: Strengths refer to positive internal features of an assessment approach or method (e.g., case-based assessment approach that authentically evaluates clinical reasoning, assessment methods with high inter-rater reliability). Weaknesses refer to internal limitations or constraints of assessment approaches or methods (e.g., memorization-focused assessment approach that limits application of knowledge, assessment methods that lack standardization between evaluators). Opportunities refer to external factors that can enhance or innovate an assessment approach or method (e.g., availability of national competency frameworks to support a competency-based assessment approach, use of new technology to streamline assessment marking). Threats refer to external challenges or pressures that could negatively impact assessment approaches or methods (e.g., accreditation requirement changes, institutional budget cuts limiting resource-intensive assessment methods).

## Dissemination plan

Dissemination will occur through multiple channels for impact across physiotherapy education and research. Findings of this scoping review are planned to be published in a peer-reviewed journal, targeting open access where possible, to maximize accessibility for educational researchers and physiotherapy educators. To support transparency, all data underlying the findings will be made available upon publishing. We will also seek to present findings at national and international conferences hosted by physiotherapy professional organizations (e.g., Canadian Physiotherapy Association Congress, World Physiotherapy Congress). Western University's Faculty of Health Sciences Knowledge Mobilization team will support broader reach by promoting publications and key findings through a coordinated social media strategy.

## Conclusion

This scoping review will provide a comprehensive synthesis of assessment approaches and methods used in physiotherapy education. The SWOT analysis of assessment approaches and methods will enable a structured and strategic

evaluation of the internal and external factors that affect assessments (e.g., implementation, effectiveness, sustainability), supporting physiotherapy programs and educators in making evidence-informed decisions about their assessment practices. Mapping assessment methods to Miller's pyramid of clinical competence will support physiotherapy educators in ensuring constructive alignment between learning outcomes, teaching and assessment methods [5,10]. Review findings may highlight under-used or innovative assessment methods that could be more widely adopted, or they could expose misalignment with competency expectations, ultimately supporting physiotherapy educators to refine or re-shape assessments to ensure educational quality. Scoping review findings will inform meaningful implementation of evidence-informed assessment strategies to optimize physiotherapy student learning and experiences and ensure graduates are well-prepared for contemporary physiotherapy practice.

## Supporting information

**S1 Appendix. Medline search strategy.**
(DOCX)

**S2 Appendix. Peer review of the electronic search strategy.**
(PDF)

**S3 Appendix. PRISMA-P checklist.**
(DOCX)

## Acknowledgments

The authors thank Meagan Stanley, MLIS, Western University, for peer review of the submitted search strategy.
All authors have approved the final version of the manuscript to be published. KK is the guarantor of the review.

## Author contributions

**Conceptualization:** Katie L Kowalski, Andrews Tawiah, Erin Miller, Greg Alcock, Heather Gillis.

**Methodology:** Katie L Kowalski, Andrews Tawiah, Erin Miller, Maren Goodman, Greg Alcock, Heather Gillis.

**Project administration:** Katie L Kowalski.

**Supervision:** Katie L Kowalski.

**Writing – original draft:** Katie L Kowalski, Andrews Tawiah.

**Writing – review & editing:** Katie L Kowalski, Andrews Tawiah, Erin Miller, Maren Goodman, Greg Alcock, Heather Gillis.

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
