## [Decision Letter · Decision Letter 0]

15 Sep 2025

Dear Dr. Kowalski,

Thank you for submitting your manuscript to PLOS ONE. After careful consideration, we feel that it has merit but does not fully meet PLOS ONE’s publication criteria as it currently stands. Therefore, we invite you to submit a revised version of the manuscript that addresses the points raised during the review process.

We look forward to receiving your revised manuscript.

Kind regards,

LS Katrina Li

Academic Editor

PLOS ONE

Journal Requirements:

3. Please amend your authorship list in your manuscript file to include author Katie Kowalski.

4. Please amend the manuscript submission data (via Edit Submission) to include author Katie L Kowalski.

Reviewers' comments:

Reviewer's Responses to Questions

**Comments to the Author**

1. Does the manuscript provide a valid rationale for the proposed study, with clearly identified and justified research questions?

Reviewer #1: Yes

Reviewer #2: Yes

2. Is the protocol technically sound and planned in a manner that will lead to a meaningful outcome and allow testing the stated hypotheses?

Reviewer #1: Yes

Reviewer #2: Yes

3. Is the methodology feasible and described in sufficient detail to allow the work to be replicable?

Reviewer #1: Yes

Reviewer #2: Yes

4. Have the authors described where all data underlying the findings will be made available when the study is complete?

Reviewer #1: Yes

Reviewer #2: No

5. Is the manuscript presented in an intelligible fashion and written in standard English?

Reviewer #1: Yes

Reviewer #2: Yes

You may also provide optional suggestions and comments to authors that they might find helpful in planning their study.

Reviewer #1: Thank you for a well written scoping review protocol that will narratively characterise the assessment approaches and methods used in physiotherapy and analyse the strengths, weaknesses, opportunities and threats of approaches and methods used.

Table 1: You may wish to spell out OSCE here or as a foot note (it is mentioned earlier in its full form in line 147, it might be worth adding "OSCE" there)

Reviewer #2: This manuscript presents a well-structured and methodologically sound protocol for a scoping review of assessment approaches and methods in physiotherapy education. The research question is clearly outlined using the PCC framework, key terms are appropriately defined, and the proposed methodology is thorough.

I was pleased to see the authors acknowledge that their review aims to determine and describe assessment approaches and methods in physiotherapy education, complemented by a SWOT analysis and, importantly, mapping of these methods to relevant competencies. As a reader, I can appreciate the importance of mapping assessment approaches to competencies, and I recognise how this alignment underpins professional standards and graduate readiness. That said, the manuscript would benefit from short justification on how the authors envisage their review might influence practice; whether by guiding educators to adapt or reshape assessment designs, highlighting under-utilised methods, or exposing misalignments with competency expectations. Making this pathway from evidence to application more explicit would sharpen the contribution of the review to educational quality.

I also note that while critical appraisal is not required for scoping reviews, some narrative consideration of study quality could add nuance to the evidence-informed outputs.

Finally, I recommend including a short dissemination plan. A clear statement on how the findings will be shared - for example, through peer-reviewed publication and conference presentations - would improve transparency and signal the intended impact of the work.

Overall, this is a carefully conceived and clearly presented protocol on a relevant and timely topic. With minor refinements to the rationale, greater clarity on anticipated implications for assessment practice, acknowledgement of study quality, and a dissemination plan, the work will be well positioned to make a meaningful contribution to physiotherapy education.

**Do you want your identity to be public for this peer review?** For information about this choice, including consent withdrawal, please see our Privacy Policy

Reviewer #1: No

Reviewer #2: **Yes: ** Emma McComb

---

## [Author Response · Author response to Decision Letter 1]

24 Sep 2025

We have incorporated feedback into the manuscript from the two reviewers, which has improved its clarity and sharpened the potential impact of study findings to physiotherapy education. We have addressed each point raised by the reviewers and responded in the submitted document titled “Kowalski et al Point-by-point response to reviewers”. The corresponding changes to the manuscript are indicated using tracked changes and uploaded with a file naming convention indicating as such.

Reviewer 1

Reviewer 1 comments Author response

Thank you for a well written scoping review protocol that will narratively characterise the assessment approaches and methods used in physiotherapy and analyse the strengths, weaknesses, opportunities and threats of approaches and methods used. Thank you for the feedback on the quality of writing.

Table 1: You may wish to spell out OSCE here or as a foot note (it is mentioned earlier in its full form in line 147, it might be worth adding "OSCE" there) We have spelled out OSCE within table 1.

Reviewer 2

Reviewer 2 comments Author response

4. Have the authors described where all data underlying the findings will be made available when the study is complete?

Reviewer 2 response: No We have added to the new dissemination plan paragraph that all data underlying findings will be made available upon publishing the review (line 253).

This manuscript presents a well-structured and methodologically sound protocol for a scoping review of assessment approaches and methods in physiotherapy education. The research question is clearly outlined using the PCC framework, key terms are appropriately defined, and the proposed methodology is thorough. Thank you for this positive feedback.

I was pleased to see the authors acknowledge that their review aims to determine and describe assessment approaches and methods in physiotherapy education, complemented by a SWOT analysis and, importantly, mapping of these methods to relevant competencies. As a reader, I can appreciate the importance of mapping assessment approaches to competencies, and I recognise how this alignment underpins professional standards and graduate readiness. That said, the manuscript would benefit from short justification on how the authors envisage their review might influence practice; whether by guiding educators to adapt or reshape assessment designs, highlighting under-utilised methods, or exposing misalignments with competency expectations. Making this pathway from evidence to application more explicit would sharpen the contribution of the review to educational quality. Thank you for this suggestion to better highlight the potential impact and use of the review findings. We have added to the discussion paragraph that review findings may highlight under-used or innovative assessment methods that could be more widely adopted, or they could expose misalignment with competency expectations, ultimately supporting physiotherapy educators to refine or re-shape assessments to ensure educational quality (line 269-272).

I also note that while critical appraisal is not required for scoping reviews, some narrative consideration of study quality could add nuance to the evidence-informed outputs. Thank you for this thoughtful point. We agree that critical appraisal is not required for scoping reviews and owing to the anticipated heterogeneity of study designs and methodologies, selecting a single critical appraisal tool to enable interpretation across included studies would be problematic. However, the data that we have planned to extract will support a narrative description of methodological features (e.g., design, reporting, sample size) that point towards study quality, allowing for a nuanced interpretation of findings. This has been added to the synthesis of results section (line 216).

Finally, I recommend including a short dissemination plan. A clear statement on how the findings will be shared - for example, through peer-reviewed publication and conference presentations - would improve transparency and signal the intended impact of the work. Thank you for this suggestion, we have added a brief dissemination plan for the review, which includes intent to publish in a peer-reviewed journal, targeting open access where possible, presenting at conferences hosted by physiotherapy professional organizations (e.g., Canadian Physiotherapy Association Congress), and a broader social media strategy through our institution’s knowledge mobilization team. This paragraph begins on line 250.

Overall, this is a carefully conceived and clearly presented protocol on a relevant and timely topic. With minor refinements to the rationale, greater clarity on anticipated implications for assessment practice, acknowledgement of study quality, and a dissemination plan, the work will be well positioned to make a meaningful contribution to physiotherapy education. Thank you for your feedback. We believe we have thoroughly addressed your points, which has added clarity and sharpened the potential impact of study findings to physiotherapy education.

---

## [Editor Report · Decision Letter 1]

8 Oct 2025

Assessment approaches and methods in physiotherapy education: A scoping review protocol

PONE-D-25-45667R1

Dear Dr. Kowalski,

We’re pleased to inform you that your manuscript has been judged scientifically suitable for publication and will be formally accepted for publication once it meets all outstanding technical requirements.

Kind regards,

LS Katrina Li

Academic Editor

PLOS ONE
---

## [Editor Report · Acceptance letter]

PONE-D-25-45667R1

PLOS ONE

Dear Dr. Kowalski,

I'm pleased to inform you that your manuscript has been deemed suitable for publication in PLOS ONE. Congratulations! Your manuscript is now being handed over to our production team.

Kind regards,

on behalf of

Dr. LS Katrina Li

Academic Editor

PLOS ONE